# Phylogenetic and Taxonomic Analyses Reveal Three New Wood-Inhabiting Fungi (Polyporales, Basidiomycota) in China

**DOI:** 10.3390/jof10010055

**Published:** 2024-01-08

**Authors:** Yang Yang, Rong Li, Qianquan Jiang, Hongmin Zhou, Akmal Muhammad, Hongjuan Wang, Changlin Zhao

**Affiliations:** 1Key Laboratory for Forest Resources Conservation and Utilization in the Southwest Mountains of China, Ministry of Education, Southwest Forestry University, Kunming 650224, China; fungiyoung@163.com (Y.Y.); hongminzhou@foxmail.com (H.Z.); 2Yunnan Key Laboratory of Gastrodia and Fungal Symbiotic Biology, Zhaotong University, Zhaotong 657000, China; 3College of Biodiversity Conservation, Southwest Forestry University, Kunming 650224, China; fungi0728pearscus@163.com (R.L.); fungiqianquanjiang@163.com (Q.J.); akmal_fungi2023@outlook.com (A.M.); 4Yunnan Forestry and Grassland Bureau, Kunming 650224, China

**Keywords:** biodiversity, ecology, molecular systematics, multiple gene, wood-inhabiting fungi, Yunnan Province

## Abstract

Three new wood-inhabiting fungal species, *Cerioporus yunnanensis*, *Perenniporiopsis sinensis,* and *Sarcoporia yunnanensis,* are proposed based on a combination of the morphological features and molecular evidence. *Cerioporus yunnanensis* is characterized by the pileate basidiomata having a fawn brown to black pileal surface, a dimitic hyphal system with clamped generative hyphae, and the presence of the fusoid cystidioles and cylindrical basidiospores (9–12.5 × 3.5–5 µm). *Perenniporiopsis sinensis* is distinct from the osseous pileus with verrucose, an orange-yellow to dark reddish-brown pileal surface with a cream margin, a trimitic hyphal system with clamped generative hyphae, and the presence of the fusiform cystidioles and ellipsoid basidiospores (9–11 × 5.5–6.5 µm). *Sarcoporia yunnanensis* is typical of the pileate basidiomata with a salmon to reddish-brown pileal surface, a monomitic hyphal system with clamped generative hyphae, and the presence of the ellipsoid basidiospores (4–5.5 × 2.5–4 µm). Sequences of ITS + nLSU + mt-SSU + TEF1 + RPB1 + RPB2 genes were used for the phylogenetic analyses using maximum likelihood, maximum parsimony, and Bayesian inference methods. The multiple genes with six loci analysis showed that the three new species nested within the order Polyporales, in which *C. yunnanensis* and *P. sinensis* nested into the family Polyporaceae, and *S. yunnanensis* grouped into the family Sarcoporiaceae.

## 1. Introduction

Fungi are eukaryotic microorganisms that play fundamental ecological roles as decomposers and mutualists of dead and living plants and animals, in which they drive carbon cycling in forest soils, mediate the mineral nutrition of plants, and alleviate the carbon limitations of other soil organisms [1,2]. Wood-inhabiting fungi form an ecologically important branch of the tree of life, based on their distinct and diverse characteristics [3]. Taxonomy and phylogeny of the Polyporales are updated continuously by mycologists with the frequent inclusion of data from DNA sequences [4,5]. In recent years, the mycologist revised the species and taxonomic status of the genus *Cerioporus* using ITS + nLSU + TEF1 datasets, while the research established the new genus *Perenniporiopsis* using ITS + nLSU + mt-SSU + TEF1 datasets and two new species have also been discovered and grouped in the genus *Sarcoporia* [6,7,8,9,10].

The genus *Cerioporus* Quél. (1886: 167), belonging to the family Polyporaceae (Polyporales, Basidiomycota), is typified by *C. squamosus* (Huds.) Quél. (1886: 167), and it is characterized by the polyporoid, chondrostereoid basidiomata with trametoid to fibroporioid habitus, hyphal system sarcomonomitic, sarcodimitic, or dimitic with arboriform sclerohyphae, generative hyphae clamped, skeletal hyphae hyaline to golden-brown, regularly branched, clavate basidia, four-spored with a basal clamp, cylindrical, navicular, fusiform or amygdaloid, and thin-walled basidiospores [10]. Based on the Index Fungorum (www.indexfungorum.org; accessed on 30 November 2023), the genus *Cerioporus* has 46 specific and registered names with 20 species that have been accepted worldwide [8,10]. The genus *Perenniporiopsis* C.L. Zhao (2017: 294), belonging to Polyporaceae (Polyporales, Basidiomycota), was typified by *P. minutissima* (Yasuda) C.L. Zhao (2017: 294), and it is characterized by pileate basidiomata, pileus solitary or imbricate, corky, becoming rigidly osseous upon drying, an orange-brown to dark reddish-brown pileal surface, hyphal system trimitic, generative hyphae hyaline, thin-walled, with clamp connections, skeletal and binding hyphae dominant, thick-walled, dextrinoid in Melzer’s reagent, oblong-ellipsoid, truncate, thick-walled, smooth, and basidiospores [9]. Based on the Index Fungorum, the genus *Perenniporiopsis* has one specific and registered name and currently one species has been accepted worldwide [9]. The genus *Sarcoporia P.* Karst. (1894: 15) belongs to Sarcoporiaceae (Polyporales, Basidiomycota), typified by *S. polyspora* P. Karst. (1894: 15), and it is typical of the annual, resupinate to effused-reflexed basidiomata, hyphal system monomitic with clamp connections, thin- to thick-walled, dextrinoid, and thick-walled basidiospores [6]. Based on the Index Fungorum, the genus *Sarcoporia* has four specific and registered names, and currently three species have been accepted worldwide [7].

Recently, the pioneering research on the *Cerioporus, Perenniporiopsis,* and *Sarcoporia* genera was significant for the molecular systematics of Polyporales [6,7,8,9,10]. The molecular phylogeny of all the polyporoid and lentinoid nodes was reconstructed using nLSU + ITS rDNA and TEF1 datasets, the data obtained from ITS + TEF + LSU coincide in support of the core Polyporaceae of ten clades corresponding to the generic level and *Cerioporus* contain generic units distinguished by polyporoid or lentinoid morphotypes, therefore, some nomenclatural innovations are given, which includes 12 synonymies of *Cerioporus* [8]. Since then, 16 new combinations of *Cerioporus* were made [10]. Phylogenetic analyses based on two datasets (ITS + nLSU and ITS + nLSU + mt-SSU + tef1) showed that specimens of *Perenniporia minutissima* (Yasuda) T. Hatt. and Ryvarden form a monophyletic well-supported clade within the core polyporoid clade, and proposed a new genus *Perenniporiopsis* [9]. Phylogenetic analyses based on two datasets (ITS + nLSU and 18S + LSU + RPB1) revealed two new species in the genus *Sarcoporia* [6,7].

Wood-inhabiting fungi are generally found in inverted wood and dead tree trunks, artificial wood products, secrete various biological enzymes that degrade the cellulose, hemicellulose, and lignin of the wood into simple inorganic substances, and play an important role in forest ecosystems as decomposers [2,3,5]. During the surveys of the wood-inhabiting fungi, we collected three new taxa of Polyporales from the Yunnan-Guizhou Plateau, China, that were not consistent with any known species. We presented the morphological characteristics and multigene molecular analyses with ITS, nLSU, mt-SSU, TEF1, RPB1, and RPB2 DNA markers that supported the taxonomy and phylogenetics of *Cerioporus, Perenniporiopsis,* and *Sarcoporia* species.

## 2. Materials and Methods

### 2.1. Sample Collection and Herbarium Specimen Preparation

Fresh basidiomata of the fungi growing on angiosperm branches were collected from the Qujing, Puer, and Honghe of Yunnan Province, China. The samples were photographed in situ and fresh macroscopic details were recorded. Photographs were taken by a Jianeng 80D camera (Tokyo, Japan). All of the photos were focus stacked and merged using Helicon Focus Pro 7.7.5 software. Specimens were dried in an electric food dehydrator at 40 °C, then sealed and stored in an envelope bag and deposited in the herbarium of the Southwest Forestry University (SWFC), Kunming, Yunnan Province, China.

### 2.2. Morphology

Macromorphological descriptions are based on field notes and photos captured in the field and laboratory and follow the color terminology of Petersen [11]. Micromorphological data were obtained from the dried specimens following observation under a light microscope [12]. The following abbreviations were used: KOH = 5% potassium hydroxide water solution, CB+ = cyanophilous, CB = cotton clue, CB– = acyanophilous, IKI = Melzer’s reagent, IKI– = both inamyloid and indextrinoid, L = means spore length (arithmetic average for all spores), W = means spore width (arithmetic average for all spores), Q = variation in the L/W ratios between the specimens studied, and *n* = a/b (number of spores (a) measured from a given number (b) of specimens).

### 2.3. DNA Extraction, PCR, and Sequencing

The EZNA HP Fungal DNA Kit (Omega Biotechnologies Co., Ltd., Kunming, China) was used to extract DNA with some modifications from the dried specimens. The ITS, nLSU, TEF1, mt-SSU, RPB1, and RPB2 regions were amplified with the ITS5/ITS4 [13], LR0R/LR7 [14], EF1-983F/EF1-2218R [15], MS1/MS2 [13], RPB1-Af/RPB1-Cf [16], and bRPB2-6F/bRPB2-7.1R [17] primer pairs, respectively. The polymerase chain reaction (PCR) procedure for ITS was as follows: initial denaturation at 95 °C for 3 min, followed by 35 cycles at 94 °C for 40 s, 58 °C for 45 s, and 72 °C for 1 min, and a final extension of 72 °C for 10 min. The PCR procedure for nLSU was as follows: initial denaturation at 94 °C for 1 min, followed by 35 cycles at 94 °C for 30 s, 48 °C for 1 min and 72 °C for 1.5 min, and a final extension of 72 °C for 10 min. The PCR procedure for TEF1 was as follows: (1) initial denaturation at 94 °C for 2.5 min, (2) denaturation at 94 °C for 45 s, (3) annealing at 60 °C for 50 s (minus 1 C per cycle), (4) extension at 72 °C for 2 min, (5) repeat for 6 cycles starting at step 2, (6) denaturation at 94 °C for 30 s, (7) annealing at 55 °C for 50 s, (8) extension at 72 °C for 1.5 min, (9) repeat for 34 cycles starting at step 6, (10) leave at 72 °C for 5 min. The PCR procedure for mt-SSU was as follows: initial denaturation at 94 °C for 2 min, followed by 36 cycles at 94 °C for 45 s, 52 °C for 45 s, and 72 °C for 1 min, and a final extension of 72 °C for 10 min. The PCR procedure for RPB1 was as follows: (1) initial denaturation at 94 °C for 2 min, (2) denaturation at 94 °C for 40 s, (3) annealing at 60 °C for 40 s, (4) extension at 72 °C for 2 min, (5) repeat for 10 cycles starting at step 2, (6) denaturation at 94 °C for 45 s, (7) annealing at 55 °C for 1.5 min, (8) extension at 72 °C for 2 min, (9) repeat for 37 cycles starting at step 6, (10) leave at 72 °C for 10 min. The PCR procedure for RPB2 was as follows: (1) initial denaturation at 95 °C for 2.5 min, (2) denaturation at 95 °C for 30 s, (3) annealing at 52 °C for 1 min, (4) extension at 72 °C for 1 min (add 1 C per cycle), (5) repeat for 40 cycles starting at step 2, (6) extension at 72 °C for 1.5 min, (7) repeat for 40 cycles starting at step 6, (8) leave at 72 °C for 5 min. The PCR products were purified and directly sequenced at Kunming Tsingke Biological Technology Limited Company, Yunnan Province, China. All of the newly generated sequences were deposited in GenBank (Table 1), and the list of known species were obtained from a previous study [5].

### 2.4. Phylogenetic Analyses

The DNA sequences were aligned in MAFFT version 7 using the G-INS-i strategy [44]. The alignment was adjusted manually using AliView version 1.27 [45]. Sequence of *Heterobasidion annosum* (Fr.) Bref. retrieved from GenBank was used as an outgroup in ITS + nLSU + mt-SSU + TEF1 + RPB1 + RPB2 (Figure 1) analysis following a previous study [5]. Sequence of *Trametes hirsuta* (Wulfen) Lloyd retrieved from GenBank was used as an outgroup in ITS (Figure 2) analysis following a previous study [27]. Sequence of *Pyrofomes demidoffii* (Lév.) Kotl. and Pouzar retrieved from GenBank was used as an outgroup in ITS + nLSU (Figure 3) analysis following a previous study [9].

Maximum parsimony (MP), Maximum Likelihood (ML), and Bayesian Inference (BI) analyses were applied to the combined three datasets. Approaches to the phylogenetic analyses process was followed by Zhao and Wu [46]. MP analysis was performed in PAUP* version 4.0b10 [47]. All of the characters were equally weighted, and gaps were treated as missing data. Clade robustness was assessed using bootstrap (BT) analysis with 1000 replicates [48]. ML was inferred using RAxML-HPC2 through the Cipres Science Gateway (www.phylo.org (accessed on 28 November 2023)) [49].

MrModeltest 2.3 [50] was used to determine the best-fit evolution model for each dataset for Bayesian inference (BI), which was performed using MrBayes 3.2.7a with a GTR + I + G model of DNA substitution and a gamma distribution rate variation across sites [51]. Four Markov chains were run for two runs from random starting trees, for eight million generations (Figure 1), one million generations (Figure 2), and 0.85 million generations (Figure 3), and trees were sampled every 100 generations.

## 3. Results

### 3.1. Molecular Phylogeny

The dataset based on ITS + nLSU + mt-SSU + TEF1 + RPB1 + RPB2 (Figure 1) comprises sequences from 146 fungal specimens representing 106 species from GenBank. The dataset had an aligned length of 7249 characters, of which 2787 characters are constant, 841 are variable and parsimony-uninformative, and 3621 are parsimony-informative. Maximum parsimony analysis yielded five equally parsimonious trees (TL = 38,172, CI = 0.2339, HI = 0.7661, RI = 0.5087, RC = 0.1190).

The dataset based on ITS (Figure 2) comprises sequences from 23 fungal specimens representing 13 species from GenBank. The dataset had an aligned length of 631 characters, of which 354 characters are constant, 42 are variable and parsimony-uninformative, and 235 are parsimony-informative. Maximum parsimony analysis yielded two equally parsimonious trees (TL = 676, CI = 0.6080, HI = 0.3920, RI = 0.6989, RC = 0.4249).

The dataset based on ITS + nLSU (Figure 3) comprises sequences from 27 fungal specimens representing 23 species from GenBank. The dataset had an aligned length of 2009 characters, of which 1555 characters are constant, 157 are variable and parsimony-uninformative, and 297 are parsimony-informative. Maximum parsimony analysis yielded 1 equally parsimonious tree (TL = 1165, CI = 0.5279, HI = 0.4721, RI = 0.5420, RC = 0.2861).

The phylogenetic tree (Figure 1) inferred from ITS + nLSU + mt-SSU + TEF1 + RPB1 + RPB2 sequences revealed that *Cerioporus yunnanensis* and *Perenniporiopsis sinensis* nested into the family Polyporaceae, and *Sarcoporia yunnanensis* clustered into the family Sarcoporiaceae. The phylogram based on the ITS gene regions (Figure 2) indicated that *C. yunnanensis* divided into genus *Cerioporus*, in which it grouped with two taxa, *C. scutellatus* (Schwein.) Zmitr., and *C. subtropicus* (B.K. Cui, Hai J. Li and Y.C. Dai) Zmitr., and then closely clustered with *C. tibeticus* (B.K. Cui, Hai J. Li and Y.C. Dai) Zmitr. Based on ITS + nLSU gene regions (Figure 3), it revealed that *P. sinensis* grouped into genus *Perenniporiopsis*, in which it was retrieved as a sister to *P. minutissima*. Based on ITS + nLSU + mt-SSU + TEF1 + RPB1 + RPB2 gene regions (Figure 1), it revealed that *S. yunnanensis* divided into genus *Sarcoporia*, in which it grouped with *S. longitubulata* Vlasák and Spirin, and then clustered with *S. polyspora* P. Karst.

### 3.2. Taxonomy

***Cerioporus yunnanensis*** Y. Yang and C.L. Zhao, sp. nov. Figure 4 and Figure 5.

MycoBank no.: 851226.

**Holotype**—China, Yunnan Province, Qujing, Zhanyi District, Lingjiao Town, Xiajia Village. GPS coordinates: 25°44′ N, 103°36′ E; altitude: 1950 m asl., on the fallen angiosperm branches, leg. C.L. Zhao, 7 March 2023, CLZhao 27270 (SWFC).

**Etymology**—***yunnanensis*** (Lat.): Referring to the locality (Yunnan Province) of the type specimen.

**Basidiomata**—Annual, pileate, odorless when fresh, hard corky when dry. Pileus applanate to triquetrous, up to 2 cm long, 1 cm wide, and 5 mm thick at base. Pileal surface fawn brown to black, distinctly sulcate, and margin obtuse. Pore surface white when fresh, becoming white to cream when dry, sterile margin distinct, up to 1 mm wide, pores round, 2–3 per mm. Context cinnamon brown to fawn brown, corky, up to 2 mm thick. Tubes cream to brown, hard corky, distinctly stratified, up to 3 mm long.

**Hyphal system**—Dimitic; generative hyphae with clamp connections, thin-walled, colorless, 2–3 µm in diameter; skeletal hyphae dominant in context, thick-walled with a narrow lumen to subsolid, colorless, 2.5–4 µm in diameter; all hyphae occasionally branched, flexuous, interwoven, IKI–, CB–, tissues unchanged in KOH.

**Hymenium**—Cystidia absent, but fusoid cystidioles present, colorless, thin-walled, 17–30 × 5–8 µm; basidia clavate, with four sterigmata and a basal clamp connection, 21–35 × 6–10 µm; basidioles in shape similar to basidia, but slightly smaller.

**Spores**—Basidiospores cylindrical, colorless, thin-walled, smooth, with oil droplets inside, IKI–, CB–, (8–)9–12.5 × 3.5–5(–6) µm, L = 10.62 µm, W = 4.30 µm, Q = 2.47–2.61 (*n* = 60/2).

**Additional specimen examined (paratype)**—China, Yunnan Province, Qujing, Zhanyi District, Xiajia Village. GPS coordinates: 25°44′ N, 103°36′ E; altitude: 1950 m asl., on fallen angiosperm branches, leg. C.L. Zhao, 6 March 2023, CLZhao 27228 (SWFC).

***Perenniporiopsis sinensis*** Y. Yang and C.L. Zhao, sp. nov. Figure 6 and Figure 7.

MycoBank no.: 851227.

**Holotype**—China, Yunnan Province, Puer, Jingdong County, Taizhong Village, Xujiaba, Ailaoshan Ecological Station. GPS coordinates: 24°23′ N, 100°53′ E; altitude: 1800 m asl., on the trunk of angiosperm, leg. C.L. Zhao, 23 August 2018, CLZhao 8315 (SWFC).

**Etymology**—***sinensis*** (Lat.): Referring to the locality (China) of the type specimen.

**Basidiomata**—Annual, pileus solitary or imbricate, osseous, without odor or taste when fresh, projecting up to 3 cm, 3 cm wide, and 1 cm thick at base. Pileal surface orange-yellow to dark reddish-brown, verrucose; margin cream, obtuse. Pore surface white when fresh, becoming pale yellowish- to yellowish-brown when dry, pores round, 4–6 per mm; Context cream, rigidly osseous, up to 7 mm thick; tubes pale cream to honey yellow, rigidly osseous, up to 3 mm long.

**Hyphal system**—Trimitic; generative hyphae with clamp connections, colorless, thin-walled; skeletal and binding hyphae colorless, thick-walled; dextrinoid in Melzer’s reagent, CB+, unchanged in KOH. In the context, generative hyphae infrequent, colorless, thin-walled, rarely branched, 1–2.5 µm in diameter; skeletal hyphae dominant, colorless, thick-walled with a distinct lumen, infrequently branched, interwoven, 3–4 µm in diameter; binding hyphae colorless, thick-walled with a narrow lumen, frequently branched, interwoven, 1–2 µm in diameter. In the hymenophoral tramal, generative hyphae infrequent, colorless, thin-walled, usually unbranched, 1–2.5 µm in diameter; skeletal hyphae dominant, colorless, thick-walled with a narrow lumen, infrequently branched, interwoven, 2–3.5 µm in diameter; binding hyphae colorless, thick-walled with a narrow lumen, frequently branched, interwoven, 1–2 µm in diameter.

**Hymenium**—Cystidia absent, but fusiform cystidioles present, colorless, thin-walled, 11–18.5 × 4.5–7 µm; basidia clavate, with four sterigmata and a basal clamp connection, 15–18 × 8.5–10 µm; basidioles in shape similar to basidia, but slightly smaller.

**Spores**—Basidiospores ellipsoid, truncate, colorless, thick-walled, smooth, dextrinoid in Melzer’s reagent, CB+, unchanged in KOH, (8.5–)9–11(–12) × (5–)5.5–6.5(–7) µm, L = 10.1 µm, W = 6.02 µm, Q = 1.62–1.68 (*n* = 60/2).

**Additional specimen examined (paratype)**—China, Yunnan Province, Puer, Jingdong County, Kongqueshan Forest Park. GPS coordinates: 24°23′ N, 100°53′ E; altitude: 1800 m asl., on fallen angiosperm branches, leg. C.L. Zhao, 22 August 2018, CLZhao 8278 (SWFC).

***Sarcoporia yunnanensis*** Y. Yang and C.L. Zhao, sp. nov. Figure 8 and Figure 9.

MycoBank no.: 851228.

**Holotype**—China, Yunnan Province, Honghe, Pingbian County, Daweishan National Forest Park. GPS coordinates: 22°57′ N, 103°42′ E; altitude: 2100 m asl., on the fallen angiosperm branches, leg. C.L. Zhao, 8 June 2020, CLZhao 18778 (SWFC).

**Etymology**—***yunnanensis*** (Lat.): Referring to the locality (Yunnan Province) of the type specimen.

**Basidiomata**—Annual, pileate, corky when fresh, brittle and hard when dry, odorless, and up to 4 cm long, 3 cm wide, and 1.5 cm thick. Pileal surface salmon to reddish-brown; margin cream, obtuse. Pore surface orange-yellow, pores angular, 2–4 per mm; context orange-brown, cottony, up to 1 cm thick; tubes pinkish-buff, up to 5 mm, extremely brittle and shattering easily when dry.

**Hyphal system**—Monomitic; generative hyphae with clamp connections, colorless, IKI–, CB–, tissues unchanged in KOH. Generative hyphae in the tube infrequent, colorless, thin-walled, easily collapsing, 1.5–3.5 µm in diameter. Generative hyphae in the context infrequent, colorless, thin-walled, 3–5 µm in diameter.

**Hymenium**—Cystidia and cystidoles absent; basidia clavate, with four short sterigmata and a basal clamp connection, 15.5–22.5 × 4–6 µm; basidioles in shape similar to basidia, but slightly smaller.

**Spores**—Basidiospores ellipsoid, colorless, thick-walled, smooth, dextrinoid, weakly cyanophilous, 4–5.5(–6) × 2.5–4(–4.5) µm, L = 4.84 µm, W = 3.31 µm, Q = 1.48 (*n* = 30/1).

## 4. Discussion

In the present study, three new species, *Cerioporus yunnanensis*, *Perenniporiopsis sinensis* and *Sarcoporia yunnanensis* are described based on phylogenetic analyses and morphological characteristics.

An outline of all genera of Basidiomycota, including three phylogenetic analyses with combined nLSU, SSU, 5.8S, RPB1, RPB2, and TEF1 datasets for the subphyla Agaricomycotina, Pucciniomycotina and Ustilaginomycotina, revealed that the genera *Cerioporus* and *Perenniporiopsis* nested into the family Polyporaceae Fr. ex Corda (Polyporales, Agaricomycetes), and the genus *Sarcoporia* clustered into the family Sarcoporiaceae (Polyporales, Agaricomycetes) [4]. In the present study, based on the ITS + nLSU + mt-SSU + TEF1 + RPB1 + RPB2 data (Figure 1), *Cerioporus yunnanensis* and *Perenniporiopsis sinensis* nested into the family Polyporaceae, while *Sarcoporia yunnanensis* clustered into the family Sarcoporiaceae, and the present results are similar to the previous topology research. Our results of the phylogram inferred from the ITS data, showed that *C. yunnanensis* grouped into *Cerioporus* (Figure 2), in which it grouped with two taxa, *C. scutellatus* and *C. subtropicus*, and then closely grouped with *C. tibeticus*. Based on the ITS + nLSU topology (Figure 3), it was revealed that *P. sinensis* was retrieved as a sister species to *P. minutissima*. Based on the ITS + nLSU + mt-SSU + TEF1 + RPB1 + RPB2 topology (Figure 1), it was revealed that *S. yunnanensis* grouped with *S. longitubulata* and *S. polyspora*. However, morphologically, *C. scutellatus* is distinct from *C. yunnanensis* by smaller pores (3–5 per mm), cyanophilous skeletal hyphae, tissues turning black in KOH and smaller basidiospores (7.8–9.2 × 3–3.6 µm) [25]; *C. subtropicus* differs from *C. yunnanensis* by smaller pores (6–8 per mm), cyanophilous skeletal hyphae, tissues turning black in KOH and smaller basidiospores (6.8–8 × 2–2.7 µm) [25]; *C. tibeticus* is separated from *C. yunnanensis* by pileal surface with color as buff, yellowish-brown or cinnamon to black from margin towards the base, ash-grey, smaller pores (4–6 per mm), and narrower basidiospores (8–10.2 × 2.5–3 µm) [25]. *Perenniporiopsis minutissima* differs from *P. sinensis* by corky pileus, wider generative hyphae (2.5–4.5 µm), wider skeletal hyphae (4–6.5 µm), longer basidia (18.5–30 × 8–13 µm), and larger basidiospores (12–15 × 6.5–8 µm) [9]. *Sarcoporia longitubulata* is separated from *S. yunnanensis* by the reddish-brown pore surface, and the circular pores [7]; *S. polyspora* differs in its soft basidiomata having the cream hymenophore, shorter basidia (9.8–16.3 × 4.1–5.4 µm), and longer basidiospores (5.2–6.1 × 2.5–3.1 µm) [52].

Morphologically, *Cerioporus choseniae* (Vassilkov) Zmitr. and Kovalenko, *C. corylinus* (Mauri) Zmitr. and Kovalenko, *C. glabrus* (Ryvarden) Zmitr., *C. leptocephalus* (Jacq.) Zmitr. and *C. melanocarpus* (B.K. Cui, Hai J. Li and Y.C. Dai) Zmitr. are similar to *C. yunnanensis* by having cylindrical basidiospores. However, *C. choseniae* differs in its wider generative hyphae (3.3–6.4 µm) and inamyloid basidiospores [53]; *C. corylinus* is separated from *C. yunnanensis* by the centrally stipitate basidiomata, cream to ochraceous pileal surface, hexagonal, bigger pores (1–2 per mm), and smaller basidiospores (6–7.5 × 2–3 µm) [8]; *C. glabrus* differs in its pale brown to corky basidiomata, isodiametric, smaller pores (7–8 per mm), and smaller basidiospores (7–9 × 2–3 µm) [10]; *C. leptocephalus* differs from *C. yunnanensis* by the centrally stipitate basidiomata, pale buff pore surface and smaller pores (7–9 per mm) [8]; *C. melanocarpus* is distinct from *C. yunnanensis* by having pale yellowish-brown context, cyanophilous skeletal hyphae, ventricose cystidioles, and narrower basidiospores (8.8–11 × 3–4 µm) [25]. *Sarcoporia neotropica* Ryvarden are similar to *S. yunnanensis* by having ellipsoid basidiospores. However, *S. neotropica* differs from *S. yunnanensis* in the soft and fleshy basidiomata, white pore surface, wider generative hyphae (3–5 µm), and smaller basidia (12–15 × 5–7 µm) [6].

In ecological and biogeographical studies, wood-inhabiting fungi are an extensively studied group of Basidiomycota, in which Polyporales species are an important group, mainly found on hardwood, although a few species grow on coniferous wood [54,55,56,57,58,59,60]. Further studies should focus on the relationships between the host and *Cerioporus*, *Perenniporiopsis,* and *Sarcoporia* species. We believe more species of Polyporales will be found in the oriental realm, since wood-inhabiting fungi are a cosmopolitan group and they are rich in the oriental realm [61,62,63,64], and it is very possible that the same phenomenon occurs for *Cerioporus*, *Perenniporiopsis,* and *Sarcoporia*.

## Figures and Tables

**Figure 1 jof-10-00055-f001:**
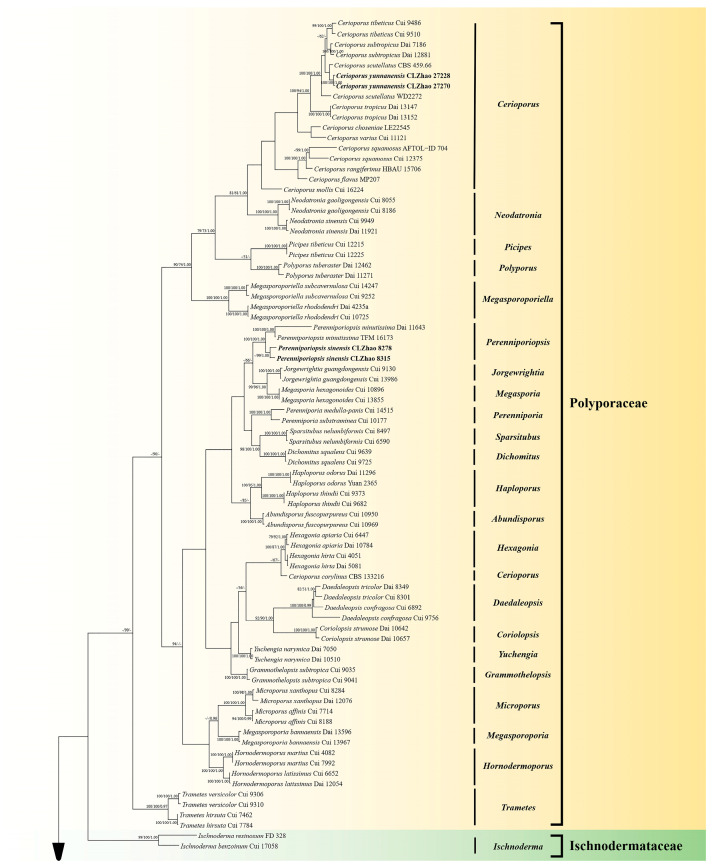
Maximum parsimony strict consensus tree illustrating the phylogeny of three new species and related species in Polyporales based on ITS + nLSU + mt-SSU + TEF1 + RPB1 + RPB2 sequences. Branches are labeled with maximum likelihood bootstrap values ≥ 70%, parsimony bootstrap values ≥ 50%, and Bayesian posterior probabilities ≥ 0.95. The new species are in bold.

**Figure 2 jof-10-00055-f002:**
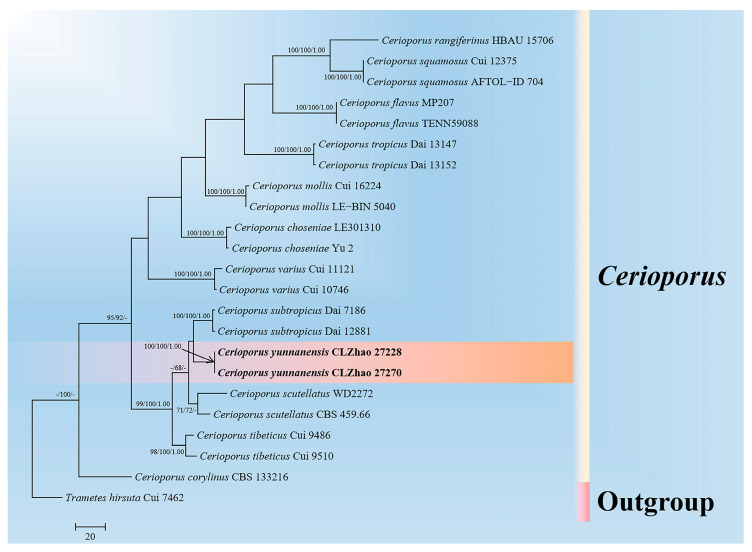
Maximum parsimony strict consensus tree illustrating the phylogeny of the new species of *Cerioporus* based on ITS sequences. Branches are labeled with maximum likelihood bootstrap values ≥ 70%, parsimony bootstrap values ≥ 50%, and Bayesian posterior probabilities ≥ 0.95. The new species are in bold.

**Figure 3 jof-10-00055-f003:**
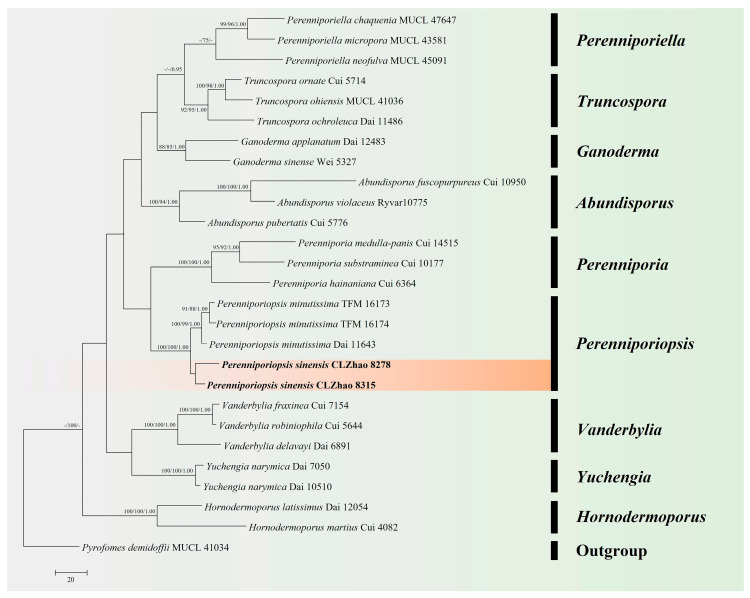
Maximum parsimony strict consensus tree illustrating the phylogeny of *Perenniporiopsis* and related species in *Perenniporia* s.l. based on ITS + nLSU sequences. Branches are labeled with maximum likelihood bootstrap values ≥ 70%, parsimony bootstrap values ≥ 50% and Bayesian posterior probabilities ≥ 0.95. The new species are in bold.

**Figure 4 jof-10-00055-f004:**
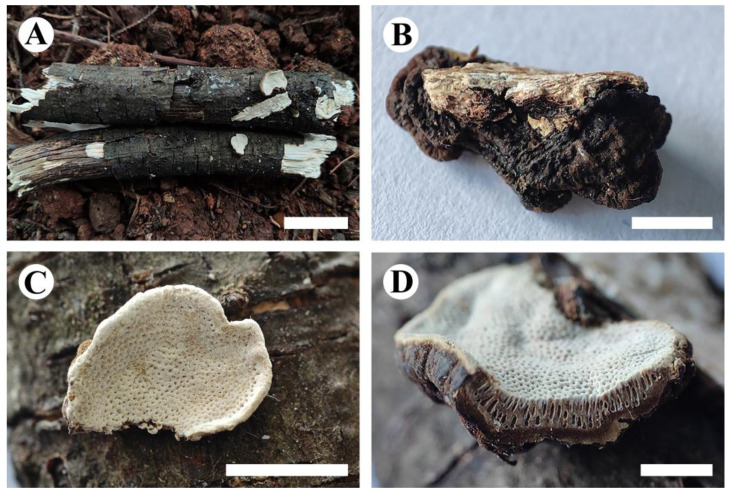
*Cerioporus yunnanensis* (holotype): basidiomata on the substrate (**A**), character hymenophore (**B**–**D**). Bars: (**A**) = 2 cm, (**B**–**D**) = 5 mm.

**Figure 5 jof-10-00055-f005:**
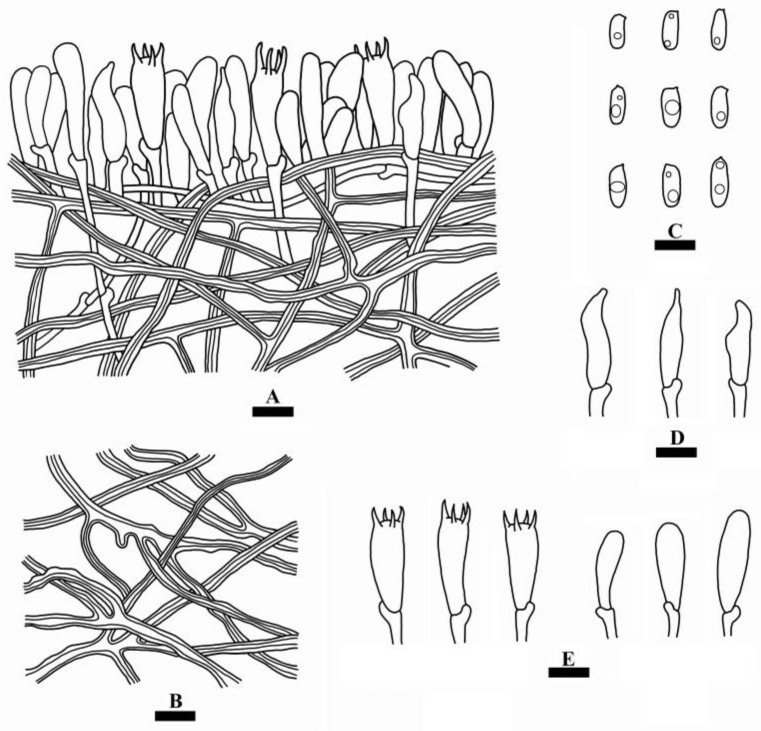
Microscopic structures of *Cerioporus yunnanensis* (holotype): a section of the hymenium (**A**), hyphae from context (**B**), basidiospores (**C**), cystidioles (**D**), basidia and basidioles (**E**). Bars: (**A–E**) = 10 µm.

**Figure 6 jof-10-00055-f006:**
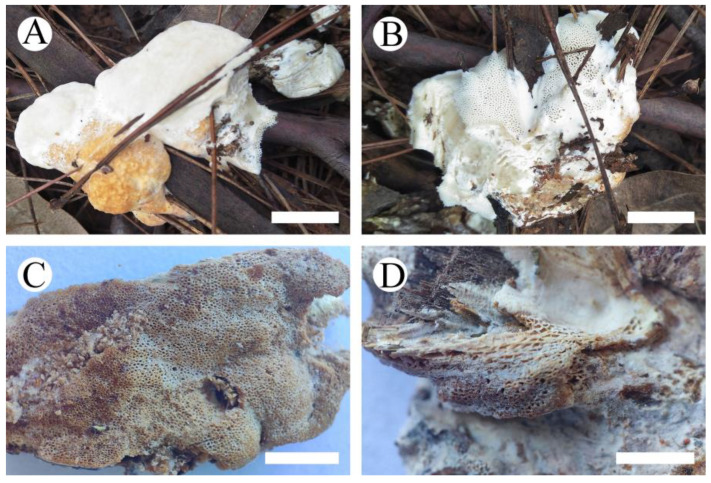
*Perenniporiopsis sinensis* (holotype): basidiomata on the substrate (**A**,**B**), character hymenophore (**C**,**D**). Bars: (**A**,**B**) = 1 cm, (**C**) = 5 mm, (**D**) = 3 mm.

**Figure 7 jof-10-00055-f007:**
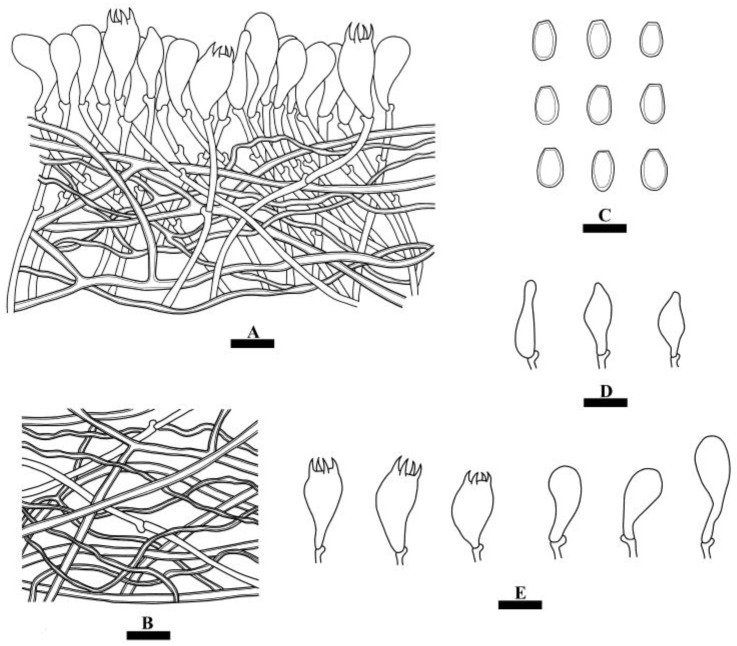
Microscopic structures of *Perenniporiopsis sinensis* (holotype): a section of the hymenium (**A**), hyphae from context (**B**), basidiospores (**C**), cystidioles (**D**), basidia and basidioles (**E**). Bars: (**A**–**E**) = 10 µm.

**Figure 8 jof-10-00055-f008:**
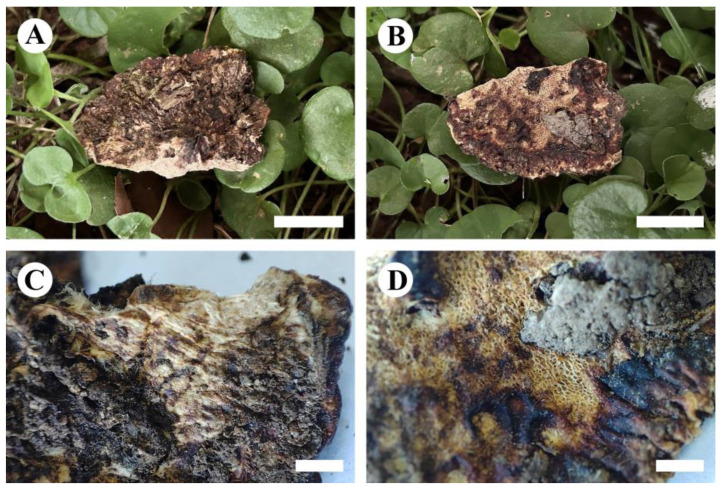
*Sarcoporia yunnanensis* (holotype): basidiomata on the substrate (**A**,**B**), character hymenophore (**C**,**D**). Bars: (**A**,**B**) = 1 cm, (**C**) = 6 mm, (**D**) = 4 mm.

**Figure 9 jof-10-00055-f009:**
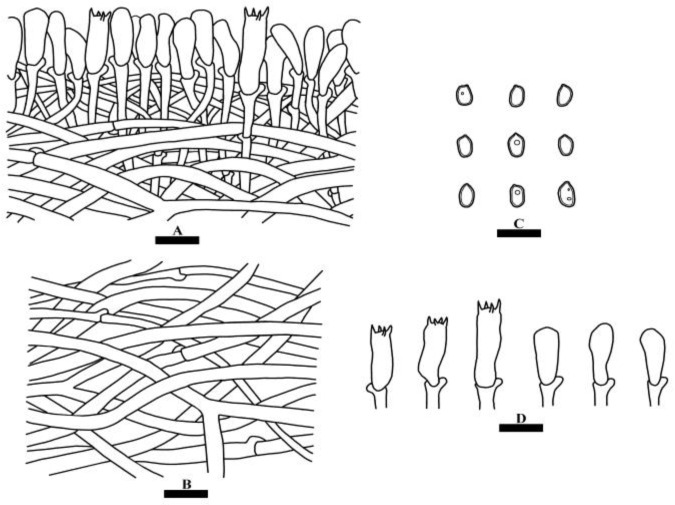
Microscopic structures of *Sarcoporia yunnanensis* (holotype): a section of the hymenium (**A**), hyphae from context (**B**), basidiospores (**C**), basidia and basidioles (**D**). Bars: (**A**–**D**) = 10 µm.

**Table 1 jof-10-00055-t001:** Names, voucher numbers and corresponding GenBank accession numbers of the taxa used in this study. The new species are in bold, NA refers to data not available.

Species Name	Sample No.	GenBank Accession No.						References
		ITS	nLSU	RPB1	RPB2	TEF1	mt-SSU	
*Abortiporus biennis*	Cui 17845	ON417149	ON417197	ON424663	ON424750	ON424821	ON417064	[5]
*Abundisporus fuscopurpureus*	Cui 10950	KC456254	KC456256	NA	NA	KF181154	KF051025	[18]
*A. fuscopurpureus*	Cui 10969	KC456255	KC456257	NA	NA	KF181155	KF051026	[18]
*A. pubertatis*	Cui 5776	KC787565	KC787572	NA	NA	NA	NA	[19]
*A. violaceus*	Ryvar10775	KF018126	KF018134	NA	NA	NA	NA	[19]
*Adustoporia sinuosa*	Cui 16253	MW377251	MW377332	MW337153	ON424752	MW337082	MW382046	[5]
*Cabalodontia delicata*	MCW 564/17	MT849295	MT849295	MT833947	NA	MT833934	NA	[20]
*Cerioporus choseniae*	LE22545	NA	KM411479	NA	NA	KM411495	NA	[8]
*C. choseniae*	LE301310	KJ595567	NA	NA	NA	NA	NA	[8]
*C. choseniae*	Yu 2	AB587632	AB587621	NA	NA	KU189924	KU189959	[21]
*C. corylinus*	CBS 133216	MK116405	NA	NA	NA	NA	NA	Unpublished
*C. flavus*	MP207	MZ997326	MZ996885	NA	NA	NA	NA	Unpublished
*C. flavus*	TENN59088	AY513571	NA	NA	NA	NA	NA	Unpublished
*C. mollis*	Cui 16224	OK642176	OK642227	NA	OK665311	OK665191	OK641964	Unpublished
*C. mollis*	LE-BIN 5040	OR683754	NA	NA	NA	NA	NA	Unpublished
*C. rangiferinus*	HBAU 15706	MZ063062	NA	NA	NA	NA	NA	Unpublished
*C. scutellatus*	CBS 459.66	MH858856	MH870495	NA	NA	NA	AF358731	[22]
*C. scutellatus*	WD2272	LC412118	AB368064	NA	AB368122	NA	NA	[23]
*C. squamosus*	AFTOL-ID 704	DQ267123	AY629320	DQ831023	DQ408120	DQ028601	NA	[24]
*C. squamosus*	Cui 12375	KX851641	NA	NA	NA	NA	NA	[21]
*C. subtropicus*	Dai 7186	JX559262	JX559301	NA	JX559310	NA	NA	[25]
*C. subtropicus*	Dai 12881	KC415183	KC415193	NA	KC415200	NA	NA	[25]
*C. tibeticus*	Cui 9486	JX559265	JX559299	NA	JX559309	NA	NA	[25]
*C. tibeticus*	Cui 9510	JX559264	JX559298	NA	JX559308	NA	NA	[25]
*C. tropicus*	Dai 13147	KC415181	KC415189	NA	KC477838	NA	NA	[25]
*C. tropicus*	Dai 13152	KC415182	KC415190	NA	KC477839	NA	NA	[25]
*C. varius*	Cui 10746	KX900013	NA	NA	NA	NA	NA	[21]
*C. varius*	Cui 11121	KX900014	KX900134	NA	NA	KX900342	KX900215	[21]
** *C. yunnanensis* **	**CLZhao 27228**	**OR771909**	**OR759766**	**OR872327**	**OR875937**	**NA**	**OR852701**	**Present study**
** *C. yunnanensis* **	**CLZhao 27270**	**OR771910**	**OR759767**	**OR872328**	**OR875938**	**NA**	**OR852702**	**Present study**
*Ceriporia lacerata*	FP-55521-T	KP135024	KP135202	KP134805	KP134915	NA	NA	[26]
*Cerrena unicolor*	He 6082	OM100740	OM083972	ON424672	ON424756	ON424825	ON417068	[5]
*Cerrena zonata*	Cui 16578	ON417153	ON417203	ON424673	ON424757	ON424826	ON417069	[5]
*Coriolopsis strumosa*	Dai 10642	JX559278	JX559303	KX885080	JX559312	KX838416	KX838379	[27]
*C. strumosa*	Dai 10657	KC867371	KC867491	KX885081	KF274650	KX838417	KX838380	[27]
*Crustoderma dryinum*	FP 105487	KC585320	KC585145	NA	NA	NA	NA	[28]
*Cymatoderma elegans*	Dai17511	ON417155	ON417205	NA	NA	NA	NA	[5]
*Dacryobolus gracilis*	He 5995	ON417156	ON417206	NA	ON424760	ON424831	ON417075	[5]
*D. sudans*	FP 101996	KC585332	KC585157	NA	NA	NA	NA	[28]
*Daedalea dickinsii*	Yuan 2685	KP171201	KP171223	NA	KR610803	KR610712	KR605982	[29]
*D. quercina*	Dai 12152	KP171207	KP171229	ON424675	KR610809	KR610717	KR605989	[5]
*Daedaleopsis confragosa*	Cui 6892	KU892428	KU892448	KU892481	KU892507	KX838418	KX838381	[27]
*D. confragosa*	Cui 9756	KU892438	KU892451	KU892483	KU892508	NA	NA	[27]
*D. tricolor*	Cui 8301	KU892426	KU892468	KU892487	KU892513	KX838423	KX838386	[27]
*D. tricolor*	Dai 8349	KU892432	KU892470	KU892490	KU892501	KX838422	KX838385	[27]
*Dichomitus squalens*	Cui 9639	JQ780407	JQ780426	KX838471	KX838478	KX838436	KX838404	[30]
*D. squalens*	Cui 9725	JQ780408	JQ780427	KX838470	NA	KX838435	KX838403	[30]
*Fibroporia albicans*	Cui 16486	OM039277	OM039177	OM037750	OM037775	OM037799	OM039212	[5]
*F. vaillantii*	Dai 23467	ON417158	ON417208	ON424680	ON424763	ON424833	ON417077	[5]
*Fomitopsis kesiyae*	Cui 16466	MN148235	OL621250	ON424688	MN158178	MN161750	OL621761	[5]
*F. pinicola*	LT 319	KF169652	NA	NA	KF169721	KF178377	NA	[31]
*Fragiliporia fragilis*	Dai 13080	KJ734260	KJ734264	NA	KJ790248	KJ790245	KJ734268	[19]
*F. fragilis*	Dai 13559	KJ734261	KJ734265	NA	KJ790249	KJ790246	KJ734269	[19]
*Ganoderma applanatum*	Dai 12483	KF494999	KF495009	NA	NA	NA	NA	[19]
*G. sinense*	Wei 5327	KF494998	KF495008	NA	NA	NA	NA	[19]
*Gelatoporia subvermispora*	Dai 22847	ON417160	ON417210	ON424695	ON424773	ON424836	NA	[5]
*Gilbertsonia angulopora*	FP 133019	KC585354	KC585182	NA	NA	NA	NA	[28]
*Grammothelopsis subtropica*	Cui 9035	JQ845094	JQ845097	NA	NA	KF181124	KF051030	[18]
*G. subtropica*	Cui 9041	JQ845096	JQ845099	NA	NA	KF181133	KF051039	[18]
*Grifola frondosa*	Dai 19172	ON417161	ON417211	ON424696	ON424774	ON424837	NA	[5]
*G. frondosa*	Dai 19175	ON417162	ON417212	ON424697	ON424775	ON424838	NA	[5]
*Haploporus odorus*	Dai 11296	KU941845	KU941869	NA	KU941916	KU941932	NA	[32]
*H. odorus*	Yuan 2365	KU941846	KU941870	NA	KU941917	KU941933	NA	[32]
*H. thindii*	Cui 9373	KU941851	KU941875	NA	KU941922	KU941938	NA	[32]
*H. thindii*	Cui 9682	KU941852	KU941876	NA	KU941923	KU941939	NA	[32]
*Heterobasidion annosum*	Dai 20962	ON417163	ON417213	ON424698	ON424776	ON529284	ON417079	[5]
*Hexagonia apiaria*	Cui 6447	KC867362	KC867481	MG867667	KF274660	MG867697	MG847228	[27]
*H. apiaria*	Dai 10784	KX900635	KX900682	MG867668	MG867677	KX900822	KX900732	[27]
*H. hirta*	Dai 5081	NA	KC867486	NA	NA	NA	NA	[27]
*H. hirta*	Cui 4051	KC867359	KC867471	NA	NA	NA	NA	[27]
*Hornodermoporus latissimus*	Cui 6652	HQ876604	JF706340	NA	NA	KF181134	KF051040	[27]
*H. latissimus*	Dai 12054	KX900639	KX900686	NA	NA	KF286303	KF218297	[27]
*H. martius*	Cui 4082	KX900640	KX900687	NA	NA	NA	KX900736	[27]
*H. martius*	Cui 7992	HQ876603	HQ654114	NA	NA	KF181135	KF051041	[27]
*Hyphoderma litschaueri*	FP 101740	KP135295	KP135219	KP134868	KP134965	NA	NA	[26]
*H. setigerum*	FD 312	KP135297	KP135222	KP134871	NA	NA	NA	[26]
*H. sordidum*	CLZhao 27390	OR141732	OR506180	OR520149	NA	OR507166	NA	[5]
*Hyphodermella rosae*	FP-150552	KP134978	KP135223	KP134823	KP134939	NA	NA	[26]
*Irpex lacteus* ^T^	FD-9	KP135026	KP135224	KP134806	NA	NA	NA	[26]
*Ischnoderma benzoinum*	Cui 17058	ON417164	ON417214	ON424699	ON424777	ON424839	ON417080	[5]
*I. resinosum*	FD 328	KP135303	KP135225	KP134884	KP134972	NA	NA	[26]
*Jorgewrightia guangdongensis*	Cui 9130	JQ314373	JQ780428	NA	NA	MG867698	KX900747	[27]
*J. guangdongensis*	Cui 13986	MG847208	MG847217	NA	MG867680	MG867699	MG847229	[27]
*Laetiporus ailaoshanensis*	Dai 13256	KF951289	KF951317	NA	KT894786	KX354625	KX354579	[33]
*L. sulphureus*	Cui 12388	KR187105	KX354486	MG867671	KX354652	KX354607	KX354560	[33]
*Laricifomes officinalis*	JV 0309/49-J	KR605821	KR605764	NA	KR610846	KR610757	NA	[29]
*Leptoporus mollis*	RLG-7163-Sp	KY948794	MZ637155	KY948956	OK136101	MZ913693	NA	[34]
*Luteoporia albomarginata*	GC 1702-1	LC379003	LC379155	LC379160	LC387358	LC387377	NA	[34]
*L. lutea*	GC 1409-1	MZ636998	MZ637158	MZ748467	OK136050	MZ913656	NA	[34]
*Megasporoporiella rhododendri*	Dai 4235a	JQ314355	KX900707	NA	KX900810	KX900841	KX900759	[27]
*M. rhododendri*	Cui 10725	KX900658	KX900708	NA	KX900811	KX900842	KX900760	[27]
*M. subcavernulosa*	Cui 14247	MG847213	MG847222	MG867673	MG867685	MG867705	MG847234	[27]
*M. subcavernulosa*	Cui 9252	JQ780378	JQ780416	MG867674	MG867686	MG867706	MG847235	[27]
*Megasporia hexagonoides*	Cui 10896	KX900651	KX900700	NA	NA	NA	KX900751	[27]
*M. hexagonoides*	Cui 13855	MG847209	MG847218	NA	MG867681	NA	MG847230	[27]
*Megasporoporia bannaensis*	Cui 13967	MG847212	MG847221	MG867672	MG867684	MG867704	MG847233	[27]
*M. bannaensis*	Dai 13596	KX900653	KX900702	NA	KX900808	KX900838	KX900754	[27]
*Metuloidea reniformis*	MCW 542/17	MT849303	MT849303	MT833950	NA	MT833940	NA	[20]
*Microporus affinis*	Cui 7714	JX569739	JX569746	NA	KF274661	NA	KX880696	[27]
*M. affinis*	Cui 8188	KX880614	KX880654	NA	NA	KX880874	KX880697	[27]
*M. xanthopus*	Cui 8284	JX290074	JX290071	NA	JX559313	KX880878	KX880703	[27]
*M. xanthopus*	Dai 12076	KX880620	KX880659	NA	KX880849	NA	KX880704	[27]
*Neodatronia gaoligongensis*	Cui 8055	JX559269	JX559286	NA	JX559317	NA	MG847236	[25]
*N. gaoligongensis*	Cui 8186	JX559268	JX559285	NA	JX559318	NA	MG847237	[25]
*N. sinensis*	Cui 9949	KX900663	KX900713	NA	NA	KX900847	KX900765	[27]
*N. sinensis*	Dai 11921	JX559272	JX559283	NA	JX559320	NA	NA	[25]
*Obba rivulosa*	Cui 16482	ON417172	ON417222	ON424712	ON424788	ON424850	NA	[5]
*Oligoporus podocarpi*	Dai 22043	MW937878	MW937885	MZ005580	MZ082977	MZ082983	MW937892	[5]
*O. rennyi*	Cui 17054	OK045508	OK045514	OK076906	OK076934	OK076962	OK045502	[5]
*Perenniporia hainaniana*	Cui 6364	JQ861743	JQ861759	NA	NA	NA	NA	[19]
*P. medulla-panis*	Cui 14515	MG847214	MG847223	NA	MG867687	MG867707	NA	[27]
*P. substraminea*	Cui 10177	JQ001852	JQ001844	NA	NA	KF181140	KF051046	[19]
*Perenniporiella chaquenia*	MUCL 47647	FJ411083	FJ393855	NA	NA	NA	NA	[35]
*P. micropora*	MUCL 43581	FJ411086	FJ393858	NA	NA	NA	NA	[35]
*P. neofulva*	MUCL 45091	FJ411080	FJ393852	NA	NA	NA	NA	[35]
*Perenniporiopsis minutissima*	Dai 11643	HQ876602	KF495015	NA	NA	KF286309	KF218303	[9]
*P. minutissima*	TFM 16173	KX962543	KX962550	NA	NA	NA	NA	[9]
*P. minutissima*	TFM 16174	KX962544	KX962551	NA	NA	NA	NA	[9]
** *P. sinensis* **	**CLZhao 8278**	**OR149913**	**OR759768**	**NA**	**OR875939**	**NA**	**OR852703**	**Present study**
** *P. sinensis* **	**CLZhao 8315**	**OR149914**	**OR759769**	**NA**	**OR875940**	**NA**	**OR852704**	**Present study**
*Phaeolus fragilis*	Cui 16579	MW377314	MW377392	NA	MW337070	MW337137	MW382095	[5]
*P. schweinitzii*	FP 133218	KC585369	KC585198	NA	NA	NA	NA	[28]
*Phanerochaete alnea*	FP-151125	KP135177	MZ637181	MZ748385	OK136014	MZ913641	NA	[34]
*rhodellum*	FD-18	KP135187	KP135258	KP134832	KP134948	NA	NA	[26]
*P. sordida*	FD-241	KP135136	KP135252	KP134833	KP134947	NA	NA	[26]
*Phanerochaetella angustocystidiata*	Wu 9606-39	MZ637020	GQ470638	MZ748422	OK136082	MZ913687	NA	[34]
*Phlebia tomentopileata*	GC 1602-67	MZ637040	MZ637244	MZ748457	OK136064	MZ913702	NA	[34]
*Picipes tibeticus*	Cui 12215	KU189755	KU189787	KU189879	KU189975	KU189902	KU189940	[21]
*P. tibeticus*	Cui 12225	KU189756	KU189788	KU189880	NA	KU189903	KU189941	[21]
*Podoscypha venustula*	Cui 16923	ON417181	ON417231	ON424722	ON424799	ON424860	ON417089	[5]
*Polyporus tuberaster*	Dai 11271	KU189769	KU189800	NA	KU189983	KU189914	KU189950	[21]
*P. tuberaster*	Dai 12462	KU507580	KU507582	NA	NA	KU507590	KU507584	[21]
*Postia hirsuta*	Cui 18347	OM039286	OM039186	NA	ON424800	OM037809	OM039221	[5]
*P. lactea*	Cui 17334	OM039287	OM039187	OM037753	OM037782	OM037810	OM039222	[5]
*Pyrofomes demidoffii*	MUCL 41034	FJ411105	FJ393873	NA	NA	NA	NA	[35]
*Radulodon casearius*	Cui 17979	ON417185	ON417236	ON424727	NA	ON424868	ON417093	[5]
*Resinoporia crassa*	H6029177	KJ028071	KT711030	NA	NA	KT711069	NA	[36]
*R. luteola*	Cui 16472	MW377319	MW377397	ON424728	MW337072	MW337140	MW382099	[5]
*Sarcoporia longitubulata*	JV 0809/8 T	KM207860	KM207863	NA	NA	NA	NA	[7]
*S. longitubulata*	JV 1009/9A	KM207861	KM207864	NA	NA	NA	NA	[7]
*S. polyspora*	Cui 16995	OM039299	OM039199	OM037761	ON424811	OM037817	NA	[5]
*S. polyspora*	Cui 17165 T	ON417192	ON417244	ON424740	ON424812	ON424878	NA	[5]
** *S. yunnanensis* **	**CLZhao 18778**	**OR771908**	**NA**	**NA**	**NA**	**NA**	**NA**	**Present study**
*Skeletocutis coprosmae*	Cui 16623	ON417193	ON417245	ON424741	ON424813	ON424879	ON417100	[5]
*S. yuchengii*	FBCC 1132	KY953045	KY953045	KY953143	NA	KY953109	NA	[37]
*Sparassis crispa*	AFTOL ID 703	DQ250597	AY629321	NA	DQ408122	NA	NA	[38]
*S. radicata*	OKM-4756	KC987580	KF053407	KY949023	NA	NA	NA	[39]
*Sparsitubus nelumbiformis*	Cui 6590	KX880632	KX880671	KX880819	NA	KX880888	KX880715	[27]
*S. nelumbiformis*	Cui 8497	KX880631	KX880670	NA	KX880856	KX880887	KX880714	[27]
*Steccherinum meridionale*	Cui 16691	ON417195	ON417247	ON424743	ON424817	ON424882	ON417102	[5]
*Trametes hirsuta*	Cui 7462	KC848299	KC848384	NA	KX880863	KX880928	KX880732	[27]
*T. hirsuta*	Cui 7784	KC848297	KC848382	NA	NA	NA	KX880731	[27]
*T. versicolor*	Cui 9306	KC848267	KC848352	NA	NA	KX880918	KX880761	[27]
*T. versicolor*	Cui 9310	KC848266	KC848351	KX880846	NA	KX880919	KX880762	[27]
*Truncospora ochroleuca*	Dai 11486	HQ654105	JF706349	NA	NA	NA	NA	[40]
*T. ohiensis*	MUCL 41036	FJ411096	FJ393863	NA	NA	NA	NA	[35]
*T. ornata*	Cui 5714	HQ654103	HQ654116	NA	NA	NA	NA	[19]
*Tyromyces chioneus*	FD 4	KP135311	KP135291	KP134891	KP134977	NA	NA	[26]
*Vanderbylia delavayi*	Dai 6891	JQ861738	KF495019	NA	NA	NA	NA	[19]
*V. fraxinea*	Cui 7154	HQ654095	HQ654110	NA	NA	NA	NA	[19]
*V. robiniophila*	Cui 5644	HQ876609	JF706342	NA	NA	NA	NA	[19]
*Wolfiporia cocos*	CBS 279.55	MW251869	MW251858	NA	MW250264	MW250253	NA	[41]
*W. hoelen*	CBK 1	KX354453	KX354689	NA	KX354685	KX354688	NA	[42]
*Wolfiporiella cartilaginea*	Dai 3764	KX354456	NA	NA	NA	NA	NA	[5]
*W. dilatohypha*	FP 94089	KC585401	KC585236	NA	NA	NA	NA	[5]
*Yuchengia narymica*	Dai 7050	JN048776	JN048795	NA	NA	KF181147	KF051053	[43]
*Y. narymica*	Dai 10510	HQ654101	JF706346	NA	NA	KF181148	KF051054	[43]

## Data Availability

Publicly available datasets were analyzed in this study. These data can be found here: [https://www.ncbi.nlm.nih.gov/ (accessed on 30 November 2023); https://www.mycobank.org/page/Simple%20names%20search (accessed on 30 November 2023)].

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
