# Peer review of "Phylogenetic and Taxonomic Analyses Reveal Three New Wood-Inhabiting Fungi (Polyporales, Basidiomycota) in China"

_jof, 2024, doi:10.3390/jof10010055_

Round 1
Reviewer 1 Report
Comments and Suggestions for Authors
The manuscript is written in a simple and concise to comprehend style. Minor changes should be done
(L 17) Cerioporus yunnanensis is characterized by the pileate basidiomata (Change in the entire manuscript)
(L 57)the genus Perenniporiopsis has one specific and registered
(L 58) name and currently one specie have been accepted worldwide
( L62)the genus Sarcoporia has four specific and registered names and currently three species have been accepted worldwide.
Author Response
Response to Reviewer 1 Comments
1. Summary |
|
|
|
||
Thank you very much for taking the time to review this manuscript. Please find the detailed responses below and the corresponding revisions/corrections highlighted/in track changes in the re-submitted files.
|
|
||||
2. Questions for General Evaluation |
Reviewer’s Evaluation |
Response and Revisions |
|||
Does the introduction provide sufficient background and include all relevant references? |
Yes |
Thank you for your evaluation. |
|||
Are all the cited references relevant to the research? |
Yes |
Thank you for your evaluation. |
|||
Is the research design appropriate? |
Yes |
Thank you for your evaluation. |
|||
Are the methods adequately described? |
Yes |
Thank you for your evaluation. |
|||
Are the results clearly presented? |
Yes |
Thank you for your evaluation. |
|||
- Point-by-point response to Comments and Suggestions for Authors
Comments 1: Page 1, line 17. Revised “Cerioporus yunnanensis is characterized by the pileate basidiocarps” as “Cerioporus yunnanensis is characterized by the pileate basidiomata” (Change in the entire manuscript).
Response 1: We have revised it according to the reviewer’s comment.
Comments 2: Page 2, line 57. Revised “the genus Perenniporiopsis has 1 specific and registered name and currently 1 species have been accepted worldwide” as “the genus Perenniporiopsis has one specific and registered name and currently one species have been accepted worldwide”.
Response 2: We have revised it.
Comments 3: Page 2, line 62. Revised “the genus Sarcoporia has 4 specific and registered names and currently 3 species have been accepted worldwide” as “the genus Sarcoporia has four specific and registered names and currently three species have been accepted worldwide”.
Response 3: We have revised it according to the reviewer’s comment.
Reviewer 2 Report
Comments and Suggestions for Authors
I did not find scientific mistakes.
I only added some remarks/corrections.

is basically ok
Author Response
Response to Reviewer 2 Comments
1. Summary |
|
|
|
||
Thank you very much for taking the time to review this manuscript. Please find the detailed responses below and the corresponding revisions/corrections highlighted/in track changes in the re-submitted files.
|
|
||||
2. Questions for General Evaluation |
Reviewer’s Evaluation |
Response and Revisions |
|||
Does the introduction provide sufficient background and include all relevant references? |
Yes |
Thank you for your evaluation. |
|||
Are all the cited references relevant to the research? |
Yes |
Thank you for your evaluation. |
|||
Is the research design appropriate? |
Yes |
Thank you for your evaluation. |
|||
Are the methods adequately described? |
Yes |
Thank you for your evaluation. |
|||
Are the results clearly presented? |
Yes |
Thank you for your evaluation. |
|||
Are the conclusions supported by the results? |
Yes |
Thank you for your evaluation. |
|||
- Point-by-point response to Comments and Suggestions for Authors
Comments 1: Page 1, line 3. Add “Chinese” before “Forest Ecological System”.
Response 1: We have revised it according to the reviewer's comment.
Comments 2: Page 1, line 23. Revised vary vocabulary, not always “is characterized”.
Response 2: We have revised it.
Comments 3: Page 1, line 28. “Polyporaceae”, families not in italics.
Response 3: We have revised it.
Comments 4: Page 1, line 33. Revised vary vocabulary, not always “grouped”.
Response 4: We have revised it according to the reviewer's comment.
Comments 5: Page 1, line 35. “Multi-genes” one word?
Response 5: We have revised it to “Multiple gene”.
Comments 6: Page 1, line 40. Add “dead and living” before “plants and animals”.
Response 6: We have revised it according to the reviewer's comment.
Comments 7: Page 2, line 54. Remove “family”.
Response 7: We have revised it.
Comments 8: Page 2, line 56. Remove “www.indexfungorum.org”, we know this from above.
Response 8: We have revised it according to the reviewer's comment.
Comments 9: Page 2, line 64. Revised “Recent” as “Recently, ....”.
Response 9: We have revised it.
Comments 10: Page 2, line 68. “characterized”, change word.
Response 10: We have revised it according to the reviewer's comment.
Comments 11: Page 3, line 96. Explain why you selected the species listed in Table 1. Which was the principle for this selection?
Response 11: We have revised it.
Comments 12: Page 6, line 124. Most species listed in Figures 1–3 were obtained from GenBank.
Response 12: We have revised it according to the reviewer's comment.
Comments 13: Page 8, line 164. “grouped”, vary vocabulary.
Response 13: We have revised it.
Comments 14: Page 14, line 316. The first sentence should be placed in the Introduction.
Response 14: We have placed it in the Introduction according to the reviewer's comment.
Comments 15: Page 14, line 318. Only white-rot fungi degrade native lignin and use it as carbon source.
Response 15: We have deleted it.